# Hooked-End Steel Fibers Affecting Damping Ratio of Modified Self-Compacting Concrete with Rubber and Marble/Granite Additions

**DOI:** 10.3390/ma17235717

**Published:** 2024-11-22

**Authors:** Amauri Ernesto Gomes, Luisa A. Gachet, Rosa Cristina C. Lintz, Mirian de L. N. M. Melo, Wislei R. Osório

**Affiliations:** 1Institute of Mechanical Engineering, Federal University of Itajubá (IEM-UNIFEI), Itajubá 37500-903, MG, Brazil; semogiro@gmail.com (A.E.G.); mirianmottamelo@unifei.edu.br (M.d.L.N.M.M.); 2Faculdade de Tecnologia, FT, Campus I, Universidade de Campinas/UNICAMP, Limeira 13484-332, SP, Brazil; gachet@unicamp.br (L.A.G.); rosacclintz@ft.unicamp.br (R.C.C.L.); 3Faculdade de Ciências Aplicadas, FCA, Centro de Pesquisa em Materiais Avançados (CPMMA), Campus II, Universidade de Campinas/UNICAMP, Limeira 13484-350, SP, Brazil

**Keywords:** mechanical properties, self-compacting concrete, waste residues, rubber content, microstructural array, composite material

## Abstract

The aim of this investigation is to determine the effects of hooked-end steel fibers on both the fresh and hardened properties of modified self-compacting concretes (SCC). For this purpose, the steel fibers are associated with other residue contents (i.e., marble, granite, and rubber). These concatenated material contents constitute a novelty since no investigations are reported. It is found that with the increase in rubber content, a better ability to absorb energy is observed. This indicates that a good alternative to structural material is provided. Fresh properties are evaluated by using flow, T_500_ time, V-funnel, and J-ring methods. The mechanical behavior is evaluated in terms of compressive strength, tensile strength, static and dynamic modulus of elasticity, and damping ratio. Experimental results of the water absorption, porosity, and density are also attained. It is also found that an SCC mixture containing steel fiber, marble/granite residue, and rubber content is a potential mixture to be considered when designing in SCC associated with an improved damping ratio. Although the rubber content decreases the mechanical behavior and slump flow, the concatenated utilization of marble/granite residues and steel fiber contents provides a slight improvement in the damping result. An environmental benefit can also be associated since cement consumption is decreased with marble additions.

## 1. Introduction

Self-compacting concrete (SCC) is able to flow and consolidate due to the effect of gravity and fill dense reinforcement efficiently, without any external vibration [1,2], has high mortar content, and uses fines and additives to maintain its cohesion. Several studies have been reported about SCC with marble and granite residues [3,4,5,6,7,8,9,10,11], SCC containing rubber [12,13,14,15,16,17,18,19,20], and SCC with steel fiber [21,22,23,24,25,26]. However, studies concatenating rubber, marble, and granite residues (MGR) with hooked-end steel fibers are scarce. The novelty intrinsically provided in this investigation concerns the fact that a new SCC is produced. This is carried out when alternative materials with distinctive nature aspects are proposed. Marble powder wastes are a by-product of the marble industry, generated in large quantities in the process of cutting and shaping marble pieces [27]. Much research has used marble and granite waste in concrete and mortar, either replacing part of the cement or sand mass or only incorporating into the mixture [28,29,30]. From the physical point of view, the presence of marble residues in the hardened cement paste has a filling effect and the porosity of the paste is decreased [31,32,33]. Marble residues are made up of inert or almost inert material [27]. Some studies show that when marble and granite residues are incorporated into the SCC (up to 10%), replacing the cement mass, there is no significant variation in the mechanical properties of concrete [5,10]. When the sand portion is replaced with ~10% wt.% granite powder, the resulting compressive strength is increased between 8% and 12% [34]. Additionally, high volumes of incorporation of marble and granite residues bring environmental impacts, minimizing CO_2_ emissions into the atmosphere, energy consumption, and natural resources [6].

Interesting properties are observed when replacing natural sand with rubber residues, such as a reduction in the slump flow of SCC [12,18], a reduction in the passing ability [35], a reduction in the concrete density [36], an improvement in the energy absorption capacity of the cement composite [37], and an increase in the ductility of the concrete [16]. Najim and Hall [38] and Li et al. [12] found that the damping rate increases with the addition of rubber residue in the concrete. Structural steel fibers are used in concrete as a reinforcement material to modify ductility, improving the toughness of hardened concrete. In the fresh state, steel fibers in the SCC reduce fluidity and hinder the ability to pass through mixing obstacles [24,39], and in the process of casting the concrete, they influence the mechanical performance of the set according to its type, its distribution and orientation [40,41,42,43].

Regarding the hardened state of the SCC reinforced with steel fibers, there is no significant change in compressive strength or in Young’s modulus [22,44,45]. Steel fibers prevent the growth of cracks in concrete, and it increases energy absorption [24,46,47]. Fiber reinforced concrete is a composite material that has improved post-cracking behavior due to the bridge that is made between the crack faces by the fibers. The majority of investigations have studied the feasibility of using alternative materials for the production of concrete.

The novelty provided in this present investigation concerns hooked-end steel fiber additions associated with rubber, marble, and granite contents. The simultaneous use of these materials on the mechanical properties of SCC is scarcely reported. Additionally, it is found that the damping ratio is improved with concatenated residue contents associated with steel fiber inclusions. It is important to remark that this investigation has certain limitations. It is anchored to specific combinations of additives and their concentrations. Evidently, some other possible interactions and effects on SCC properties cannot be focused on these attained results when a broader range of construction scenarios is considered. Another limitation concerns the fact that the attained results are laboratory-based analyses. This could limit the direct translation of the findings to real-world applications without comprehensive field validation. Thus, further empirical research to fully harness the benefits of the modified SCC in various environmental and operational contexts is requested.

## 2. Experimental Procedures

### 2.1. Materials

High early strength (HES) Portland cement (according to ASTM C150) is used, and its physical properties and chemical composition are shown in Table 1 and Table 2. The fine aggregate consists of natural quartzite sand; the coarse aggregate is of basaltic origin (Table 1). Silica fume is the mineral addition used in the SCC preparation. This material is incorporated into the mixture intending to increase the resulting strength. This is due to the effects of micro filler and refinement of the pore structure and cement hydration products. Table 1 and Table 2 show the physical properties and chemical composition. Marble and granite residue (MGR) is used in the production of SCC as the fine content. It is obtained in a wet condition from the ornamental stone industry, located at Limeira, SP, Brazil, and used in a dry condition. The physical properties and chemical composition of the MGR are listed in Table 1 and Table 2. The Laser Method (ISO 13320:2009) is applied in order for the granulometric characterization to be carried out. Rubber waste tires are also used, as depicted in Figure 1. 

The physical properties and the chemical elements are demonstrated in Table 1 and Table 2. Potable tap water is used in the molding and curing stages. A polycarboxylate ether-based superplasticizer (SP) is used, as shown in Table 1.

Hooked-end steel fibers (HESF), designated as Dramix^®^ RC65/35B, are used, as depicted in Figure 2. The tensile strength of the steel fibers is 1345 (±50) N/mm^2^, the length is 35 (±1) mm, the diameter is 0.55 (±0.05) mm, the form factor is 65, and the modulus of elasticity is 200,000 N/mm^2^.

It is remarked that no straight steel fiber results are included and compared with the attained results in the present investigation. This is due to, firstly, the intercept mechanism and inhibition of crack growth reducing the likelihood of further crack propagation are different. Additionally, this is reasonably reported in the literature, which will negatively impact the novelty of this investigation. 

### 2.2. Production of Concrete 

After the wheel design, six distinctive SCC mixtures are proposed. The purpose is to demonstrate the effects of the MGR content associated with HESF and the rubber in both the fresh and hardened states of the SCC samples examined. The selection of mixes is based on dosage studies for SCC developed by [11,12,30,48,49]. The content of silica fume used in the mixtures is 10%; concerning the cement mass, the water/cement ratio is 0.58, and the mortar content is close to 65%. Considering the fiber consumption utilized in this study, both 10 (±1) kg/m^3^ and 20 (±1) kg/m^3^ are used, similarly previously provided in [30]. The dosages adopted for fibers are based on previously reported studies [24,30,50,51].

Table 3 shows the mixtures of the studied concretes. In order to guarantee reproducibility and to determine the compressive strength, for each one of the proposed mixtures, at least 6 specimens are molded and tested at 7 and 28 days of age. Thus, the number of the used specimens to conduct the compression, tensile, and modulus of elastic measurements are 10, 4, and 4, respectively. This totals 60 + 24 + 24 = 108 specimens, considering all proposed mixtures.

To determine Young’s moduli and damping of distinct proposed samples, 06 specimens are elaborated. A concrete mixer at an environmental temperature of about 20 (±5) °C is used in order for the concrete mixing to proceed. 

The molding stage is very similar to that carried out in our previous investigation, in which no fiber contents are concatenated with MGR and rubber contents [30]. For this reason, no repeated sequence of this similar experimental stage is again related [30]. However, it is important to remember that the water-to-cement ratios applied in this study, specifically concerning the SCC/20SF/30MGR and SCC/20SF/30MGR/5R samples, are slightly lower. 

### 2.3. Fresh State and Hardened State Measurements

The properties of the SCC specimens in the fresh state are evaluated according to the Brazilian standard ABNT NBR 15823-1:2017, which has an international equivalence standard. Slump flow, flow time, and visual stability index tests (Abrams cone method) are carried out, as described in ABNT NBR 15823-2:2017. Both J-ring and V-funnel methods are also described in ABNT NBR 15823-3:2017. The experimental procedure to obtain or evaluate these properties is also detailed in our previously reported articles [30].

Compressive strength tests are carried out utilizing cylindrical specimens 100 *×* 200 (±1) mm according to ABNT NBR 5739:2018. The tensile strength is determined using cylindrical specimens utilizing a dimetrical compression method (ABNT NBR 7222:2011) and modulus of elasticity by the compression (ABNT NBR 8522:2017). More details are also described in our previously reported manuscript [30]. Concerning the dynamic moduli of elasticity, these are determined, as well as the damping rate, by the Impulse Excitation Technique (ASTM C215:2014 and ASTM E1876:2015). This constitutes a non-destructive test in which the moduli of elasticity are measured from the natural frequencies of the vibration of the sample with a regular geometry (cylindrical samples 100 × 200 mm, ±1 mm). The acoustic tests are carried out according to the arrangement shown in Figure 3a. The specimens are supported on two steel cables, with tension and adjustable positions on the rigid support (manufactured by ATCP—Sonelastic^®^, ATCP Physical Engineering 735A Leda Vassimon, 14026-567, Ribeirão Preto, Brazil; www.atcp-ndt.com, accessed on 24 April 2024) for bars and cylinders. This support is in accordance with the requirements of ASTM E1876:2015.

In this study, the steel cables are positioned at a distance of 0.224 × L from their ends, where L is the length of the specimen to simulate the specific condition of the free–free contour. A specially developed hammer, with a rubber handle and a steel ball at the end, is used to apply manual external excitation. The rubber cable has appropriate dynamic characteristics, such as very low natural frequencies and high damping, so as not to interfere with test measurements. The steel sphere with a diameter of 12.7 (±0.2) mm proved to be very effective in providing sufficient impact energy to excite the frequencies in the range of interest. A microphone (directional acoustic sensor CA-DP with specific technical characteristics designed by enterprise ATCP Physical Engineering) to capture the acoustic response transmitted by the surface of the specimen is used. The dedicated software made it possible to adjust acquisition parameters, such as sampling rate, filtering, and windowing so that natural frequencies are adequately identified and extracted with precision. 

From the acoustic response caused by the short-term mechanical impact on the specimen and, based on its mass, geometry, and dimensions, the dynamic modulus of elasticity is calculated, as shown in Figure 3a. The damping is calculated using the logarithmic decrement method (Figure 3b), which consists of the ratio between two successive amplitudes of the signal. For this purpose, a viscoelastic damping model is considered [52]. The response time of an oscillatory system with a degree of freedom, with viscous damping when excited by an impulse, is described in Equation (1):(1)$$z(t)=z{(e}^{-{\zeta \omega }_{0}t})\sin(\omega_{d}t)$$
where *z* (μm), *ω*_0_ and *ω_d_* (expressed in Hz) represent the natural frequency of vibration and damped natural frequency, as prescribed in Equation (2).


(2)
$$\omega_{d}=\omega_{0}\sqrt{1-\zeta^{2}}$$


Considering the responses at moments *t = t_n_* (enesimal time, seconds) represented by A (μm) and *t* = *t_n_* + 2π *r*/*ω_d_*, with *r* being period, expressed by $A_{n}$ (μm), Equation (3) is obtained (Figure 3b): 


(3)
$$\frac{A_{n}}{A}=\exp\left(-\zeta \frac{\omega_{0}}{\omega_{d\;}}2\pi n\right)=\exp\left[-\frac{\zeta }{\sqrt{1-\zeta^{2}}\;}2\pi n\right]$$


Therefore, the logarithmic decrement (δ) is obtained as follows in Equation (4):(4)$$\delta =\frac{1}{n}\ln(\frac{A}{A_{n}})=\frac{2\pi \zeta }{\sqrt{1-\zeta^{2}}}$$
and the damping factor (ζ) is obtained by Equation (5):(5)$$\zeta =\frac{1}{\sqrt{\zeta =\frac{1}{\sqrt{1+({2\pi /\delta )}^{2}}}1+({2\pi /\delta )}^{2}}}$$

When the damping is small (ζ < 0.1), the damping frequency is almost equal to the natural frequency, that is, $\omega_{d}$ ≅ *ω*_0_, and then Equation (3) is reworked, as shown in Equation (6):
(6)$$\frac{An}{A}≅\exp(-\zeta 2\pi n)$$
or, as expressed by Equation (7) when ζ < 0.1, as follows:


(7)
$$\zeta =\frac{1}{2\pi }\ln(\frac{A}{A_{n}})=\frac{\delta }{2\pi }$$


The dynamic deformation modulus is approximately equal to the initial tangent modulus determined in the static test [53,54]. This relationship is not easily determined based on physical behavior, as the heterogeneities of the two moduli are differently affected, as previously reported [55].

Some empirical expressions relating to the static (Es) and dynamic (Ed) moduli were reported (both expressed in MPa) at BS 8110-2:1985 [55]. Considering the concrete with a cement content of less than 500 kg/m^3^ or concrete with normal density aggregates, the static modulus, in MPa, is expressed by Equation (8).
E_S_ = 1.25 × E_d_ – 19 (8)

Lyndon and Baladran [56] have reported Equation (9) to describe the E_S_ as follows:E_S_ = 0.83 × E_d_(9)

On the other hand, Popovics [57] has reported Equation (10) as follows:E_S_ = κ × E_d_^1.4^ × ρ^−1^
(10)
where ρ is the specific mass (kg/m^3^) of the concrete and κ is a constant that depends on the units of measurement.

**Figure 3 materials-17-05717-f003:**
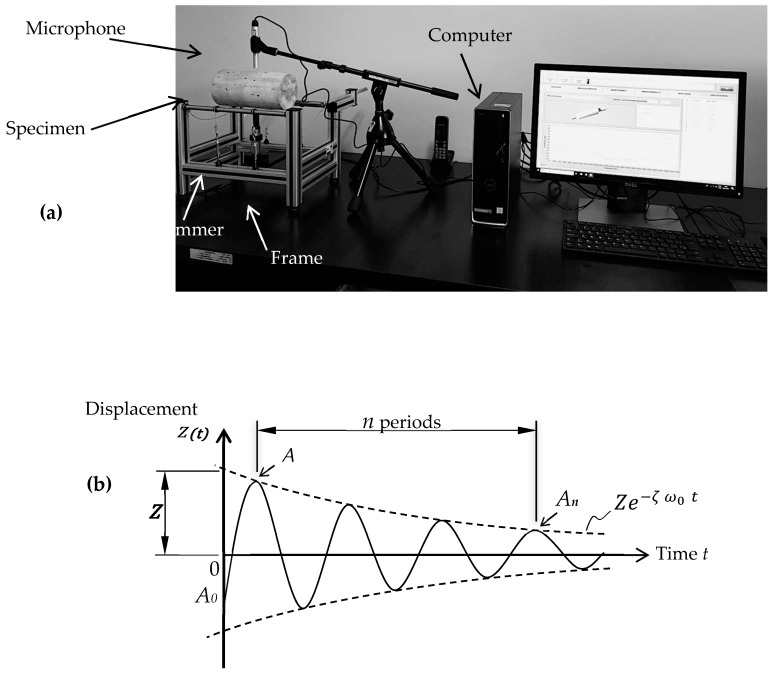
(**a**) Photograph representing the dynamic testing by impulse stimulus (damping and dynamic elastic modulus) and (**b**) response to the impulse of a damped system with a degree of freedom 0 < ζ < 1 at time *t* = 0, adapted from [58,59].

### 2.4. Microstructural Characterizations

In order to characterize the examined samples that were prepared using distinctive mixtures, a scanning electron microscope, SEM (TESCAN^®^ model VEGA3, Brno, Czech Republic) coupled with energy-dispersive x-ray spectroscopy (EDS), is utilized. These procedures are also mentioned in our previous paper [30]. All samples are withdrawn from the representation regions, which are intended to be characterized. Duplicates are considered to provide reproducibility of the attained analyses. It is remarked that gold sputtering (model MEV LEO 430i, Zeiss, USA) is used to prepare the surface sample for each specific condition. The majority of observations are obtained using electron beam energy of 20 kV, beam current of 500 pA, and a working distance of ~19 mm.

## 3. Results and Discussion

### 3.1. Fresh State Properties 

The experimental results of the fresh state of the examined SCC samples are shown in Table 4. With respect to the spreading test, all the concretes produced in this study are in accordance with the specifications of the Brazilian standards of ABNT NBR 15823-1:2017. A minimum limit attained value of 550 (±2) mm and a maximum of 850 (±2) mm for slump flow is required. It is observed that the mix containing rubber shows a decreased spreading value for both the groups with 10 kg/m^3^ and 20 kg/m^3^ of steel fibers contents. For the SCC/20SF/30MGR/5R sample, a lower spreading value than other examined samples is attained. This seems to be associated with a higher fiber content (20 kg/m^3^) and a greater amount of rubber waste. Considering the visual stability index, neither segregation nor exudation is observed. These results of the fresh state are similar to those previously published [14,16,34,53].

Based on the experimental results of the V-funnel, fluidity, and viscosity in the samples containing rubber content, i.e., designated as the SCC/10SF/30MGR/2.5R and SCC/20SF/30MGR/5R samples, higher flow time results are observed than the groups with 10 kg/m^3^ and 20 kg/m^3^ steel fiber contents. All the examined SCC samples are classified as VF 1. This means that the reached value is less than or equal to 8 s. Based on the J-ring results, all mixtures show values equal to or less than 25 (±0.5) mm. This shows that a good ability to pass through obstacles is attained, and a classification PJ 1 is indicated. 

The parameter T500 is recognized as a flow test providing the time required for concrete to flow and spread inside a 500 circle. Using this result, it is possible to classify the mix as VS1, except the SCC/20SF/30MGR/5R sample, which is classified as VS2 due to the higher rubber and steel fiber contents (20 kg/m^3^). Bušic et al. [35] also observed that the passing ability is reduced with the rubber residue addition.

It is worth noting that the concatenated use of the HESF + rubber + marble and granite residues in SCC has not previously been reported. Since its properties in the fresh state are satisfactory, the properties in the hardened state are determined.

### 3.2. Hardened State Properties

Table 5 shows the experimental results obtained in the tests in the hardened state of the SCC. Considering the results of the compressive strengths, the obtained values reveal that values higher than 34 (±2) MPa at 7 days are attained. Since HES cement is used, it is recognized that a more rapid hydration is provided. Additionally, higher cohesion than conventional cement (ordinary Portland) is also reached [60,61].

It is recognized that all studied concrete compositions are commonly classified as structural concrete according to ABNT NBR 8953:2015. Except for the sample with 5% rubber (SCC/20SF/30MGR/5R), all other samples indicate the group II class, which has compressive strength ≥55 MPa, which is considered high-strength concrete, at 28 days. In a previous investigation by Aïtcin [62], it is classified as high-performance concrete, i.e., class I (between 50 and 75 MPa).

Regarding the addition of the MGR content into the mixture, a small increase in the compressive strength of the SCC/20SF/30MGR mixture is observed when the SCC/20SF sample is compared. The MGR consumption of 105 kg/m^3^ for the SCC/20SF/30MGR mixture evidences that no substantial modification is verified, even with the decrease in cement consumption (350 kg/m^3^). This seems to be compensated by the MGR content, resulting in a slight increase (of about 2.5%) in the compressive strength when the SCC/20SF sample is compared. This correlates with reducing the pores of the cement paste since the MGR portion fills these voids, providing more cohesion of the paste and increasing the packaging of the concrete.

However, the values obtained for the SCC/10SF and SCC/10SF/30MGR samples are very close. The MGR consumption of 105 kg/m^3^ for the SCC/10SF/30MGR sample did not affect the compressive strength when the SCC/10SF sample was compared. This occurred probably due to the sand portion being replaced with the MGR content. It is interesting that a cement consumption of 366 kg/m^3^ is maintained.

The compressive strengths of the group with HESF consumption of 10 kg/m^3^ (SCC/10SF, SCC/10SF/30MGR, SCC/10SF/30MGR/2.5R) showed higher values than the group with HESF consumption of 20 kg/m^3^ (SCC/20SF, SCC/20SF/30MGR, SCC/20SF/30MGR/5R). This indicated that the increase in the fiber rate has no effect on this property, according to previous studies [22,23,24].

For the sample of the group with HESF consumption of 10 kg/m^3^, the cement consumption is kept constant (366 kg/m^3^), and the sand portion is replaced with both the rubber and MGR residue contents. This provided a compressive strength higher than the group with 20 kg/m^3^ of HESF and cement consumption of 365 kg/m^3^ (the SCC/20SF sample), 350 kg/m^3^ (the SCC/20SF/30MGR sample), and 345 kg/m^3^ (the SCC/20SF/30MGR/5R sample).

It is also found that the compressive strength is strongly affected in mixtures containing rubber residue, i.e., the SCC/10SF/30MGR/2.5R and SCC/20SF/30MGR/5R samples. The lowest compressive strength is that of the SCC/20SF/30MGR/5R sample. It corresponds to a decrease of about 36% when the SCC/20SF sample is compared. The SCC/10SF/30MGR/2.5R mixture showed a 14.5% reduction in compressive strength when compared with the SCC/10SF sample. The decrease in compressive strength with the increase in rubber content is also previously reported [16,36,38,63,64,65]. This occurrence is attributed to two main reasons: (a) firstly, the cracks are rapidly initiated at neighboring rubber particles and concrete paste, and (b) the rubber particles are weakly adhered to the paste, behaving like voids in the concrete matrix [16,64].

Regarding the tensile strength, this property is less affected than the compressive strength for all mixtures examined. Considering the rubber compositions, a decrease of about 13% is verified for the SCC/10SF/30MGR/2.5R sample when the SCC/10SF are compared, and similarly when the SCC/20SF/30MGR/5R and the SCC/20SF samples are also compared. In previous studies, this behavior was also reported [16,23,63,64]. From this point, it is important to remark that the decision for rubber content to be adopted in the proposed mixture is examined in the present investigation. Although it is recognized that rubber potentially decreases compressive behavior, certain improvements in tensile strength can be attained [60,61,63,64,65]. Based on this previous perspective, the rubber contents were considered to prepare other mixtures containing rubber contents. 

Table 6 shows that at 28 days of age, the static moduli of elasticity of the SCC/20SF and SCC/20SF/30MGR samples are very similar. Similarly, this also occurs when the SCC/10SF and the SCC/10SF/30MGR samples are compared. Comparing the mixtures without residues, the SCC/20SF and SCC/10SF samples revealed a decrease or difference of about 2.5%. When the rubber compositions are considered, the decreases are ~22% for the SCC/20SF/30MGR/5R when compared to the SCC/20SF sample and about 6% for the SCC/10SF/30MGR/2.5R sample when the SCC/10SF sample is considered.

Aslani and Kelin [23] found that with an increase in the steel fiber content in SCC, an increase in the elasticity modulus is attained. On the other hand, Hilal [16] also found that the static elastic modulus decreased with the increase in the rubber content, similar to that observed for both the tensile and compressive strengths. Based on the fact that the dynamic longitudinal modulus is lower than the flexional, the longitudinal dynamic modulus is considered. The non-destructive method (IET) presents values of the longitudinal dynamic elasticity moduli higher than the static moduli at 28 days of age, as shown in Table 6 and Figure 4a.

This fact is associated with the IET method, which consists of determining the natural vibration frequencies of the concrete specimen in the free-free condition through excitation by a slight mechanical impact. This mechanical excitation applied to the specimen produces very low stress levels during the determination of these parameters; therefore, the formation of microcracking and creep effects is not provided. For these reasons, it can be considered that the dynamic modulus is associated only with the elastic phenomena of the material and is closer to the initial tangent modulus. This corresponds to the elastic behavior of the concrete obtained at the beginning of the stress/strain curve [53,54]. The values of dynamic moduli are usually higher than static moduli due to the period of time characteristic of the vibration used in dynamic methods is regularly less than one millisecond (ms). With this, the occurrence of anelastic mechanisms with a relaxation time higher than 1 ms is prevented [66].

The IET method has some advantages when compared to the static method. The non-destructive test has reproducibility in a short period using the same sample. Therefore, a smaller number of samples is required, and there is less susceptibility to experimental errors due to the number of variables provided [67]. 

Figure 4b shows the results of empirical data related to the attained static (Ec) and dynamic (Ed) and compared with previously reported results [56,57,58], and prescribed into BS 8110-2:1985 [56]. It is observed that in the experimental values of dynamic (Ed) and static moduli (Es) of the SCC compositions examined, the following correlation is attained, i.e., Es = 0.896 × Ed^1.0^ associated with R^2^ = 0.99. It is observed that the experimental values overestimate those previously reported [55,56,57]. The variability in the results is justified by the multiphase nature of the concrete, which influences the mechanisms that it deforms.

Figure 4c shows the correlation considering an exponential dependence of the tensile property with the compressive strength at 7 days (expressed by TS = 0.38 × CS ^0.6^) and at 28 days (expressed by TS = 0.71 × CS ^0.5^). Khatri et al. [68] and Xavier et al. [30] reported similar correlations when silica fume/furnace slag/fly ash and MGR contents were used, respectively. The equations proposed in the present study for both the experimental results at 7 and 28 days are located between the upper and lower limits proposed. 

The experimental results at 28 days of compressive strength as a function of porosity are shown in Figure 5. The Ryshkewitch’s equations are described, CS = 58exp (−0.082 P), and the experimental results at 28 days, CS = 211.21 exp (−0.111 P). Similar trends are observed.

Table 6 shows the values obtained in the experimental tests of the damping factor of the SCC samples examined. The damping factor of the studied SCC is higher than steel and cast iron. The damping rate of the steel is reported between 0.001 and 0.002 [69], and the corresponding value of the cast iron is about 0.0023 [70]. Regarding the SCC/20SF sample (with consumption of HESF being 20 kg/m^3^) and the SCC/10SF (with HESF consumption of 10 kg/m^3^), it is observed that with the increase in HESF consumption, the damping factor is about 9% increased, as shown in Table 6. However, there is a decrease in this property for the SCC/20SF/30MGR and SCC/10SF/30MGR samples when the SCC/20SF and SCC/10SF samples are compared, i.e., about 6% and 2%, respectively. 

With regard to the SCC/20SF/30MGR/5R and the SCC/10SF/30MGR/2.5R samples, a slight increase of 1.34% is observed when the SCC/20SF/30MGR/5R and the SCC/20SF samples are compared. When the SCC/10SF/30MGR/2.5R sample is compared with the SCC/10SF samples, the observed difference is about 2.6%.

Considering those samples containing both MGR content and rubber residue portions, i.e., 2.5 and 5%, the two highest reached values (0.35% and 0.38%, respectively) of damping factors are associated with these mentioned samples, i.e., designated as the SCC/10SF/30MGR/2.5R and the SCC/20SF/30MGR/5R samples. This suggests that a better energy dissipation capacity is provided when the modified concrete is subjected to a dynamic load, as previously reported [12,38,71].

In the rubber portion, when incorporated into the concrete, certain voids in the paste are prevalent, and consequently, the resulting porosity is increased. The experimental results of the specific mass, void index, and water absorption by immersion, measured according to ABNT NBR 9778:2009, are shown in Table 7. The observation of the attained values shown in Table 7 clarifies that the HESF and rubber contents provide the highest values of both the water absorption and voids (porosity).

Figure 6a shows the influence of porosity in relation to the damping rate. Thakare et al. [71] also found that the air entrapped in fresh mortars increased with rubber fiber incorporation. Based on these previous assertions and attained experimental observations, it is deduced that in an SCC containing rubber residue, the porosity increases, associated with a decrease in the compressive strength, the tensile strength, and the dynamic modulus. On the other hand, it is clarified that the damping appreciably increased.

Li et al. [12] concluded that the SCC damping rate increases linearly with the rubber content. This suggests that a better energy dissipation capacity is attained when subjected to dynamic load. Najim and Hall [38] also found that the damping coefficient increases with the increase in rubber content in the mixture.

Based on the aforementioned results, it is interesting to analyze the concatenated effects of the MGR and HESF contents on the two main important properties of an SCC, i.e., slump (fresh property) and compressive strength (hardened property). Additionally, considering the fact that, under certain determined conditions, damping should also be considered, both slump flow and damping are evaluated. The experimental variations of the slump flow and damping ratio as a function of the compressive strength are demonstrated in Figure 6b. This is due to slump flow and damping seems to be competitive properties mainly when the MGR and HESF contents are increased. 

Considering that the design of a mixture requires compressive strength in a magnitude between 70 and 75 MPa, the HESF content has no substantial effect on the slump flow results. However, the MGR content clarifies that slump flow can be slightly improved; mainly, the HESF portion is increased. This is observed when the same magnitude of compressive strength (between 70 and 75 MPa) is considered. Considering this same range of compressive behavior, the damping results reveal that the 30 MGR content has a deleterious effect. Interestingly, the HESF content demonstrates a positive effect on the damping. 

For instance, the SCC/20SF has a damping considerably higher (~9%) than the SCC/10SF sample. Similarly, when analyzing the SCC/10SF/30MGR and SCC/20SF/30MGR samples, the damping results reveal that, although 30MG content is present in both mixtures, the sample with higher steel fiber content (i.e., SCC/20SF/30MGR) evidenced higher damping (~7%) than other one. On the other hand, when the SCC/20SF is compared with the SCC/20SF/30MGR and the SCC/10SF with the SCC/10SF/30MGR, the effect of the MGR content is indicated. It reveals a deleterious effect of the MGR content on the damping results, i.e., it decreases by about 6%. Additionally, when a lower range of compressive strength is acceptable in a certain design mixture, for instance, between 40 and 50 MPa, it is deduced that the slump flow and damping ratio are competitive properties. The highest damping ratio is that of the SCC/20SF/30MGR/5R samples, while their corresponding slump flow result is the lowest attained result. The increase in damping is intimately associated with 20SF content, and the considerable decrease (~20%) in their slump flow is attributed to rubber content, as shown in Figure 6b. It was previously reported that rubber content, depending on some limited characteristics and conditions, in a general way, provides a deleterious effect on mechanical behavior [60,61,62,63,64,71,72,73,74,75,76]. It was also reported that the steel fiber content (up to certain limits) improves the damping behavior [77], as also observed in this study.

When the SCC/20SF/30MGR/5R and the SCC/10SF/30MGR/2.5R samples are compared, the SF content increases the damping results while rubber content decreases the slump flow. Associated with this, a considerable and substantial decrease in compressive strength (~38%) is observed. This is also associated with double rubber content in the SCC/20SF/30MGR/5R sample, as depicted in Figure 6b. Summarizing, it is worth noting that the rubber addition induces a certain worsening in the compressive strength of concrete. 

In the next section, the micrographs of the reinforced concrete using the SEM technique demonstrate that the concrete with rubber content has various microcracks. It is also shown that HESF has an interface with cement paste without “voids”, as will be described and discussed.

### 3.3. SEM Micrographs of the Examined SCC

Figure 7 shows the microstructure of the SCC samples examined under different magnifications. Comparing Figure 7b and Figure 7d with the mixtures depicted in Figure 7a,c, it is noted that the compositions with marble and granite residues (SCC/10SF/30MGR and SCC/20SF/30MGR) result in their corresponding cement pastes being denser and more consolidated with aggregates and fibers than other ones. The hexagonal plates of calcium hydroxide (CH) and calcium silicate hydrate (C-S-H) particles are also characterized. Commonly, a higher amount of calcium and silica induces the substantial formation of C-S-H gel in a mixture. In addition, these associated residues increase the consolidation of the microstructure. These constituted particles are finer than the aggregates. Consequently, the voids are filled, and the packaging is improved, providing a compact structure, and both the quantity and the size of the pores are decreased. This corroborates with a lower void index and water absorption, confirming the experimental values obtained.

It is also found that no interfacial transition zone (ITZ) is clearly characterized. The interaction at the interface between steel fiber and cement paste seems to provide a corrosion layer formation. This migrates to cement paste, forming a corrosion-filled paste region without ITZ constitution, as previously reported [30,78,79]. All these assertions working together seem to be responsible for the observed gain in the mechanical behavior, as demonstrated in Table 5 and 6. This behavior was also previously observed [11,30]. Typical SEM micrographs of the SCC/10SF/30MGR/2.5R (with 2.5% rubber residues) and the SCC/20SF/30MGR/5R samples (with 5% rubber residues) are shown in Figure 8a,b. An open structure with crack propagation is depicted. A clear, characterized transition zone, with an increase in the volume of large voids, is also shown. This has occurred due to the incorporation of the rubber residue and entrapped air. It is remembered that the highest void index and water absorption is that of the examined sample, as demonstrated in Table 7. 

Associated with these values, lower compressive and tensile strengths and modulus of elasticity are observed. Additionally, the highest damping factor is also observed, as shown in Table 5 and Table 6. The same characteristics are verified in relation to the microstructure corresponding with the rubberized concretes, as previously reported [72,73,74,75,76].

## 4. Conclusions

Based on the attained experimental results of the modified and reinforced concretes containing distinct steel fiber contents and associated with MGR and rubber particle additions, the following conclusions can be drawn. 

Distinct portions of the hooked-end steel fibers, rubber residues, and marble and granite residues are successfully added to the constituted reinforced concretes in order to attain substantial improvements in the damping results.It is found that marble and granite residue additions provide a high-density microstructural array. The observed particles are much thinner than the aggregates, and the voids are filled. Consequently, an increase in the paste density and the packaging of the particles associated with increasing the mechanical properties is achieved.It is corroborated that the marble and granite portions are used to minimize the loss of mechanical properties due to the incorporation of the rubber particles. The concrete with rubber residues showed a higher level of porosity than other ones. This affected the transition zone, and the mechanical behavior decreased. On the other hand, a higher capacity to absorb energy and, consequently, a higher damping factor than other examined samples is obtained. Additionally, it is also remarkable that the rubber content addition relatively worsens the compressive strength of concrete.When designing an SCC mixture intending to improve the resulting damping ratio, associated with a compressive strength range between 70 and 75 MPa, the steel fiber addition has no demonstrated negative effects on the slump flow results. Although the rubber content decreased the mechanical behavior and slump flow, the concatenated utilization of the MGR and hooked-end steel fiber can be considered in designing an SCC. This is mainly in order to reach slight improvements in the damping results under certain acceptable compressive strength conditions. With this, the SCC/20SF/30MGR/5R sample seems to be a potential mixture, which also reveals a potential environmentally friendly aspect due to the decreasing trend of cement consumption when incorporating marble.It is worth noting that in a construction project (e.g., flooring, underground, or precast), there are a great variety of distinctive other fiber types, e.g., flat end, undulated, hooked flat-end fiber, and hooked glued fiber (ArcelorMittal^®^). The differences among these are geometry, dimensions, and corresponding tensile strengths, i.e., 1100, 1500, 1200, and 2400 MPa, respectively. Based on this, it is clearly perceived that flat fibers (quasi-conventional straight) have lower tensile strength than hooked (~1350 MPa). This considerably affects the resulting mechanical properties, depending on the desired construction type. Thus, the decision to adopt a conventional fiber or a hook or other type depends strongly on the desired requirements, as well as the other additives utilized and demonstrated in the present investigation.

## Figures and Tables

**Figure 1 materials-17-05717-f001:**
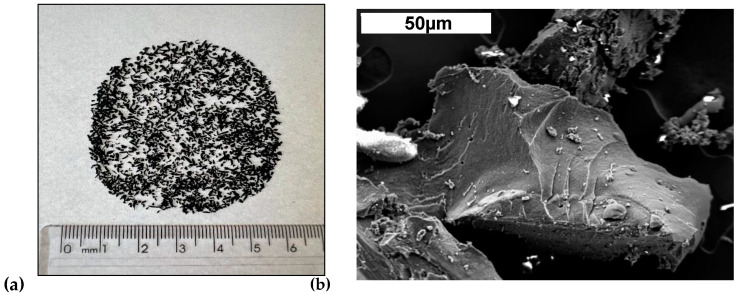
Rubber residues (**a**) macroscopic scale and (**b**) secondary electron microscope (SEM) image using the secondary electron (SE) technique to characterize rubber particles.

**Figure 2 materials-17-05717-f002:**
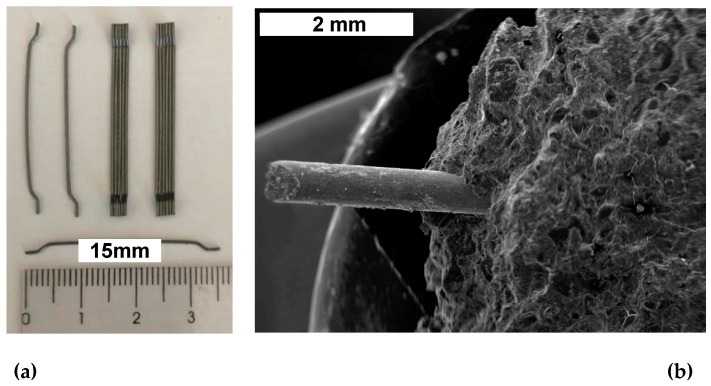
(**a**) Hooked-end steel fibers (HESF) designated as Dramix^®^ RC65/35B and (**b**) typical SEM/SE image evidencing steel fiber reinforced concrete sample.

**Figure 4 materials-17-05717-f004:**
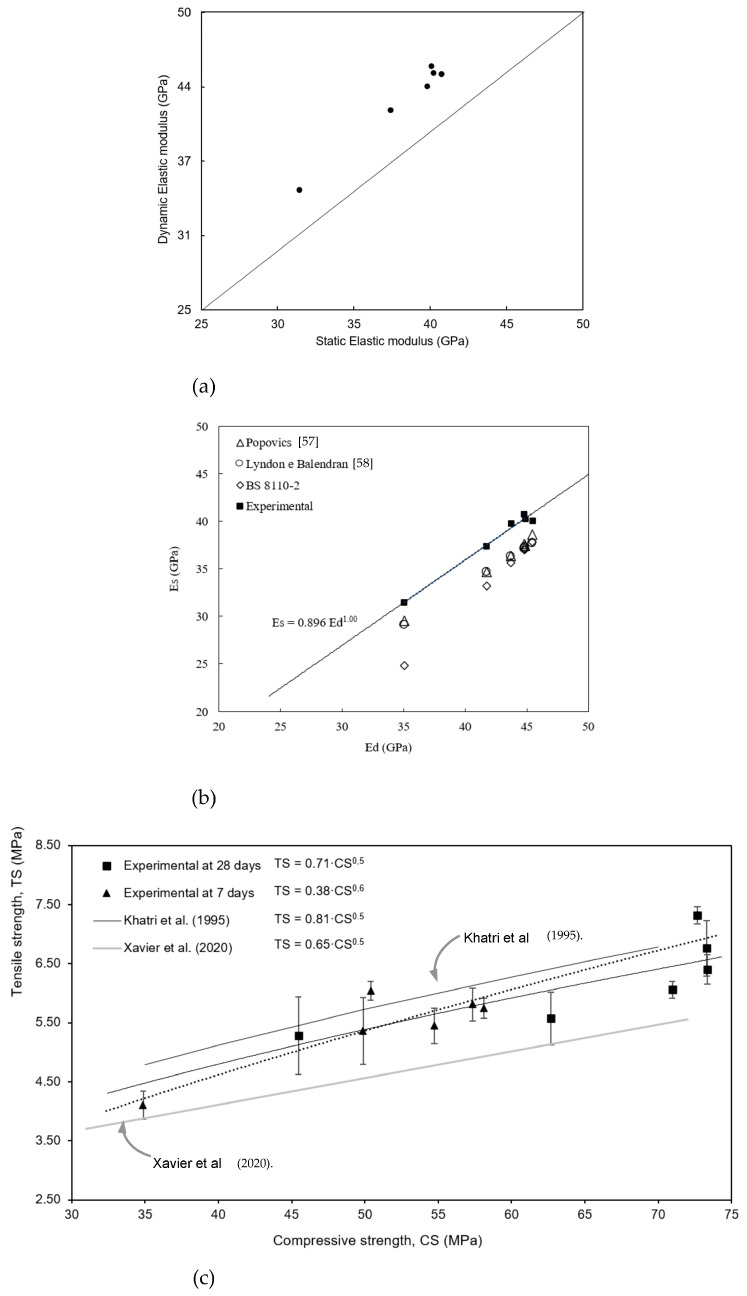
(**a**) Relation between dynamic modulus and static modulus of the SCC samples, (**b**) considering some empiricals concerning static modulus and dynamic modulus, and (**c**) tensile-to-compressive strength ratios.

**Figure 5 materials-17-05717-f005:**
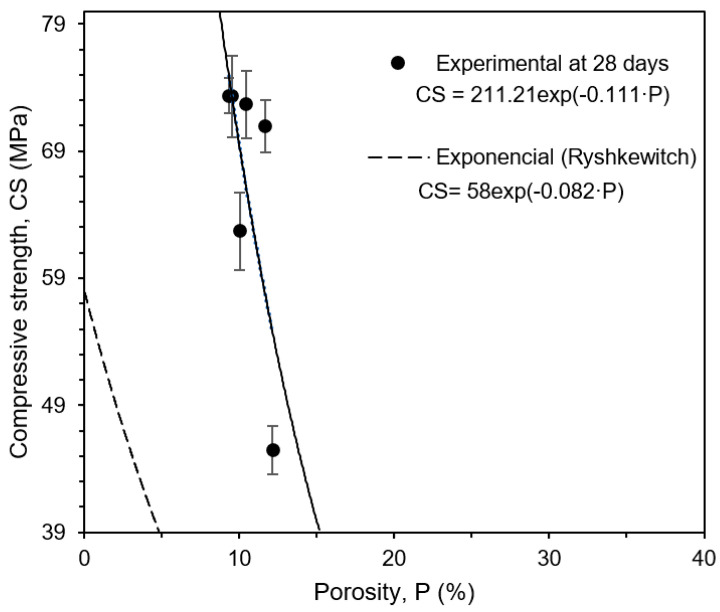
Ryshkewitch’s equation describes the compressive strength (CS) as a function of porosity (P), considering 28 days of curing.

**Figure 6 materials-17-05717-f006:**
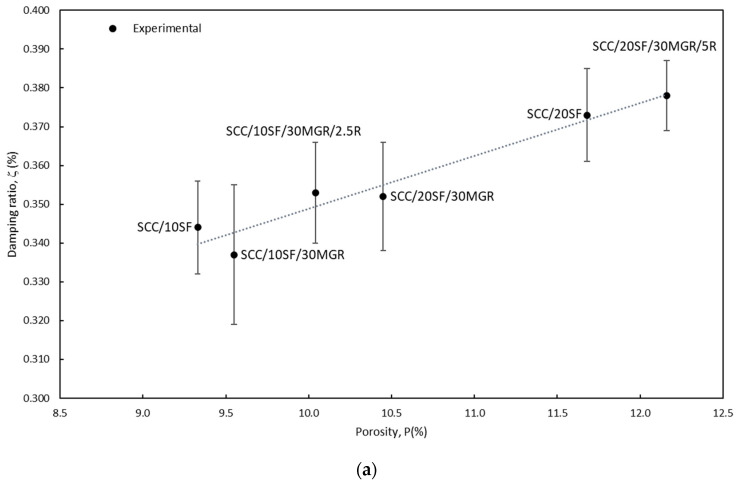
(**a**) Experimental damping ratio as a function of the porosity level, and (**b**) reveals damping ratio and slump flow variations with compressive strengths for all examined concrete samples.

**Figure 7 materials-17-05717-f007:**
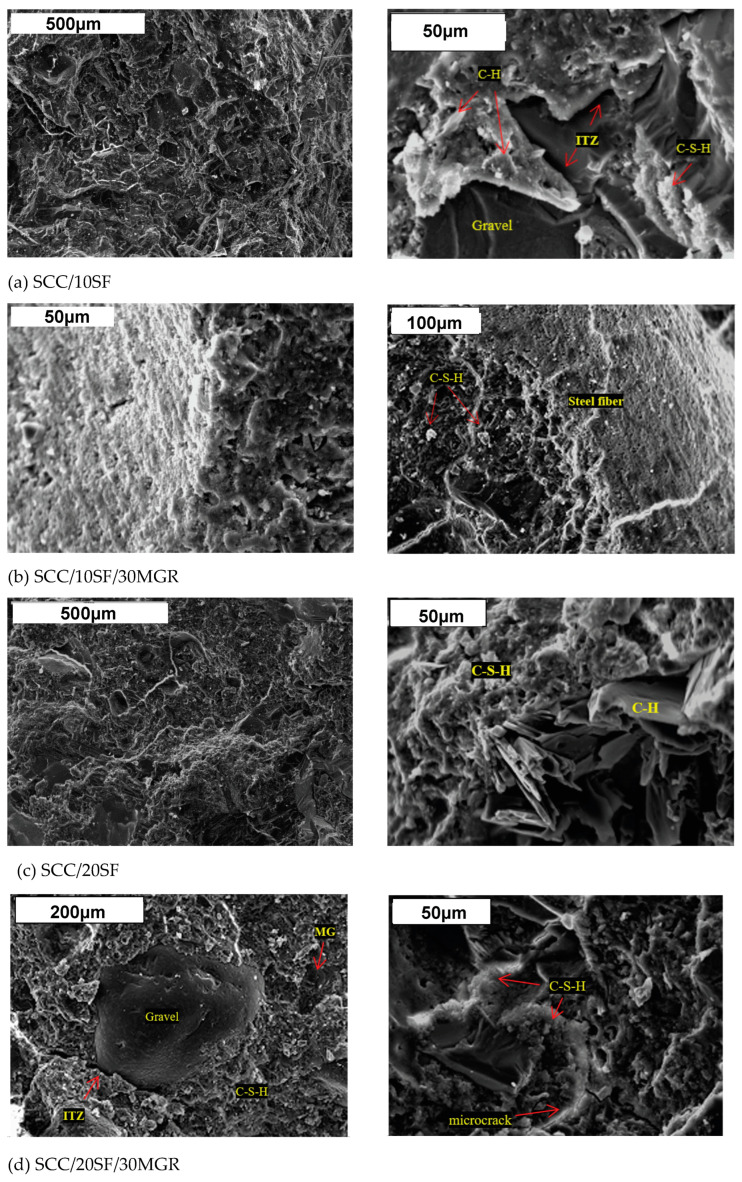
SEM (SE technique) images of concrete mixes with distinct steel fiber contents and MGR: (**a**) the SCC/10SF, (**b**) the SCC/10SF/30MGR, (**c**) the SCC/20SF, and (**d**) the SCC/20SF/30MGR samples.

**Figure 8 materials-17-05717-f008:**
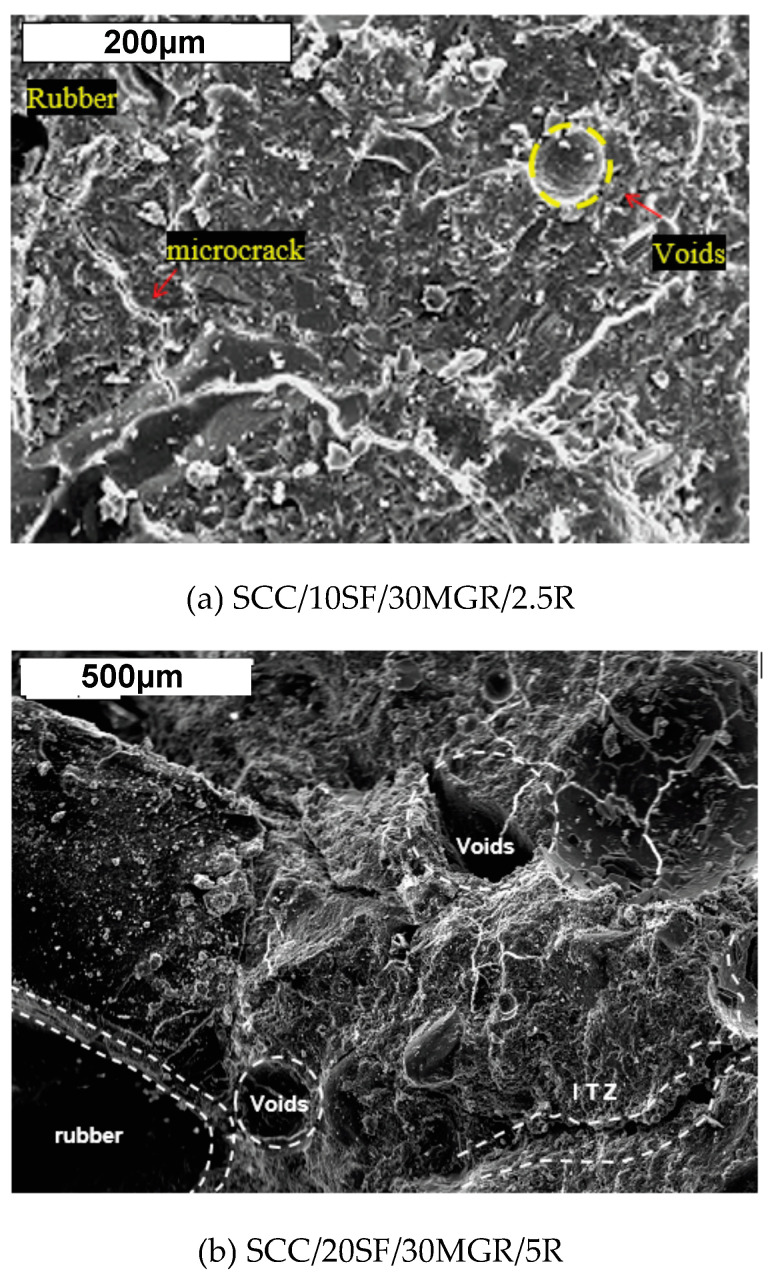
SEM (SE technique) images of concrete mixes with distinct steel fiber contents, MGR, and rubber: (**a**) the SCC/10SF/30MGR/2.5R sample and (**b**) the SCC/20SF/30MGR/5R sample.

**Table 1 materials-17-05717-t001:** Physical characterization of the materials used in the composition of the concretes.

Physical Properties	Cement	Silica Fume	Fine Aggregate	Coarse Aggregate	Marble/Granite (MGR)	Rubber Residues	SP
Density (g/cm^3^)	3.08	2.21	2.65	3.01	2.58	1.16	1.08
Unit weight (g/cm^3^)	1.03	-	1.50	1.51	-	0.39	-
Max diameter (mm)	-	-	1.20	9.50	0.10	2.40	-
Fineness modulus	-	-	1.76	5.47	10.28	2.82	-
Water absorption (%)	-	-	0.20	1.40	-	-	-

**Table 2 materials-17-05717-t002:** Chemical compositions of cement, silica fume, MG, and rubber. All values vary up to 10% from these absolute values.

	Cement	Silica Fume	Marble/Granite (MG)	Rubber Residue
Chemical composition (%)				
CaO	63.33	0.36	14.3	-
SiO_2_	19.19	95.61	48.2	-
Al_2_O_3_	5.15	0.17	12.0	-
Fe_2_O_3_	2.8	0.08	5.13	-
MgO	0.92	0.55	2.82	-
Na_2_O	-	0.19	2.03	-
K_2_O	-	1.29	3.77	-
TiO_2_	-	-	1.17	-
Lost on ignition	8.13	1.75	10.58	-
Insoluble residue	0.48	-	-	-
Chemical element (%)				
C	-	-	-	91.5
Zn	-	-	-	3.5
O	-	-	-	3.3
S	-	-	-	1.2
Na	-	-	-	0.2
H	-	-	-	0.2
Ca	-	-	-	0.1

**Table 3 materials-17-05717-t003:** Concrete mixture proportions, results of the fresh and hard states tests of the mixtures. All values range up to 10% from these absolute values.

	SCC/10SF	SCC/10SF/30MGR	SCC/10SF/30MGR/2.5R	SCC/20SF	SCC/20SF/30MGR	SCC/20SF/30MGR/5R
Cement (kg/m^3^)	366	366	366	365	350	345
Silica fume (kg/m^3^)	37	37	37	37	35	35
MG (kg/m^3^)	0	110	110	0	105	104
Rubber (kg/m^3^)	0	0	9	0	0	17
Sand (kg/m^3^)	1036	922	904	1033	991	976
Coarse aggregate (kg/m^3^)	761	761	761	759	728	718
Steel fiber (kg/m^3^)	10	10	10	20	20	20
Water (kg/m^3^)	212	212	212	212	203	200
Superplasticizer (%)	2	2	2	2	2	2

**Table 4 materials-17-05717-t004:** Properties in the fresh state of the SCC. Error values are up to 10%.

	SCC/10SF	SCC/10SF/30MGR	SCC/10SF/30MGR/2.5R	SCC/20SF	SCC/20SF/30MGR	SCC/20SF/30MGR/5R
Slump flow (mm)	730	837	725	735	725	595
T_500_ (s)	1.10	1.86	1.40	1.00	2.00	3.00
J-ring (mm)	20.00	25.00	5.50	17.50	15.75	6.25
V Funnel (s)	3.10	3.18	4.20	8.00	4.10	8.00

**Table 5 materials-17-05717-t005:** Experimental results of hardened states of the examined samples at 7 and 28 days.

Sample	Compressive Strength, CS (MPa)		Tensile Strength, TS (MPa)	
	7 Days	28 Days	7 Days	28 Days
SCC/10SF	58.1 (±1)	73.3 (±1.4)	5.7 (±0.2)	6.4 (±0.2)
SCC/10SF/30MGR	57.3 (±1)	73.3 (±3.2)	5.8 (±0.3)	6.8 (±0.5)
SCC/10SF/30MGR/2.5R	49.9 (±1.5)	62.7 (±3.1)	5.4 (±0.6)	5.6 (±0.4)
SCC/20SF	50.0 (±2)	71.0 (±2.0)	6.0 (±0.2)	6.1 (±0.1)
SCC/20SF/30MGR	55 (±2)	72.7 (±2.6)	5.7 (±0.3)	7.2 (±0.1)
SCC/20SF/30MGR/5R	34.9 (±2.7)	45.5 (±1.9)	4.1 (±0.2)	5.3 (±0.7)

**Table 6 materials-17-05717-t006:** Experimental results of the damping ratio and the moduli of elasticity determined by static and dynamic using IET of the examined SCC samples at 28 days.

Sample	Static Modulus (Gpa)	Dynamics Flexural Modulus (Gpa)	Dynamics Longitudinal Modulus (Gpa)	Damping Ratio ζ (%)
SCC/10SF	39.8 (±1.2)	44.4 (±1.6)	43.7 (±1.3)	0.34 (±0.01)
SCC/10SF/30MGR	40.2 (±1.7)	45.6 (±0.2)	44.9 (±0.6)	0.34 (±0.02)
SCC/10SF/30MGR/2.5R	37.4 (±1.6)	42.5 (±0.7)	41.7 (±0.6)	0.35 (±0.01)
SCC/20SF	40.7 (±1.9)	45.5 (±0.5)	44.8 (±0.5)	0.37 (±0.01)
SCC/20SF/30MGR	40.1 (±0.6)	45.9 (±0.5)	45.4 (±0.3)	0.35 (±0.01)
SCC/20SF/30MGR/5R	31.4 (±1.2)	35.1 (±0.6)	35.0 (±0.5)	0.38 (±0.01)

**Table 7 materials-17-05717-t007:** Experimental results of density, water absorption, and void index tests.

	SCC/10SF	SCC/10SF/30MGR	SCC/10SF/30MGR/2.5R	SCC/20SF	SCC/20SF/30MGR	SCC/20SF/30MGR/5R
Water absorption (%)	4.02 (±0.08)	4.08 (±0.06)	4.40 (±0.01)	5.04 (±0.57)	4.53 (±0.03)	5.80 (±0.12)
Void index (%)	9.33 (±0.17)	9.55 (±0.09)	10.04 (±0.08)	11.68 (±1.27)	10.45 (±0.07)	12.16 (±0.21)
Specific mass (kg/m^3^)	2320 (±3.48)	2339 (±17.05)	2280 (±15.66)	2319 (±12.58)	2306 (±3.48)	2097 (±12.04)

## Data Availability

All research data supporting this publication are directly available within this publication.

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
