# Peer review of "Hooked-End Steel Fibers Affecting Damping Ratio of Modified Self-Compacting Concrete with Rubber and Marble/Granite Additions"

_materials, 2024, doi:10.3390/ma17235717_

Round 1
Reviewer 1 Report
Comments and Suggestions for Authors
A. Objectives
This investigation delves deeply into the effects of incorporating hooked-end steel fibers within self-compacting concretes (SCC), which are further modified by adding rubber and marble/granite waste. This study aims to explore a pioneering blend of materials to potentially elevate both fresh and hardened properties of SCC. By interweaving varied components such as steel fibers, marble, granite, and rubber residues into the concrete matrix, this research seeks to unveil a composite material that stands out for its enhanced energy absorption capabilities and superior damping ratios. The overarching goal is to assess the viability of these innovative SCC mixtures for broader structural applications, thus pushing the boundaries of traditional concrete performance while maintaining a keen eye on sustainability and environmental benefits through the reuse of industrial by-products.
B. Methodology
The methodology employed in this research is comprehensive, involving a multi-faceted experimental approach to thoroughly assess the properties of the modified SCC. The investigation begins with evaluating the fresh properties of the concrete mixtures, employing techniques such as flow, T500 time, V-funnel, and J-ring tests to gauge workability and flowability. To delve into the mechanical prowess of these novel concrete mixtures, a series of rigorous tests are conducted, focusing on key metrics such as compressive strength, tensile strength, static and dynamic modulus of elasticity, and, notably, the damping ratio. The scope of the study extends to encompass an analysis of water absorption, porosity, and density, providing a holistic view of the material characteristics. Central to this methodology is exploring mixtures enriched with steel fiber, marble/granite residues, and rubber content, examining their collective potential to revolutionize SCC design principles with an emphasis on improved damping ratios and sustainability.
C. Results
The study's findings are revealing, showcasing the beneficial impact of rubber content on enhancing the material's ability to absorb energy, thus positing rubber-enriched SCC as a promising candidate for structural material applications. Furthermore, the investigation highlights that SCC mixtures imbued with steel fibers, marble/granite residues, and rubber content not only promise an improved damping ratio but also do so without drastically affecting the slump flow or undermining the mechanical integrity of the concrete. This delicate balance suggests a significant environmental upside, primarily through the reduction in cement consumption achieved by incorporating marble additions, thus presenting a compelling case for the environmental and mechanical viability of these modified SCC mixtures.
D. Discussion
In the discussion, the intricate balance between enhanced damping properties and the preservation of mechanical integrity within SCC mixtures is thoughtfully explored. The study critically examines the interplay among the unique components of the modified concrete, shedding light on the potential for groundbreaking advancements in concrete design, particularly for applications where energy absorption and damping are paramount. This section elaborates on the implications of each material's addition, dissecting how these ingredients collectively contribute to the composite material's overall performance. The discussion sets the stage for future explorations into optimizing concrete mixtures, leveraging the novel insights gained from this study to push the envelope in sustainable construction practices.
E. Originality
By venturing into relatively uncharted territory, this study makes a seminal contribution to the field of concrete technology. The originality of the research lies in its exploration of a unique composite material that integrates hooked-end steel fibers with rubber and marble/granite residues within SCC. This innovative approach not only promises enhanced mechanical properties and damping ratios but also aligns with sustainable construction practices by repurposing industrial waste. The study's pioneering blend of materials offers fresh perspectives on the possibilities of concrete technology, setting a new benchmark for developing advanced, sustainable, and mechanically robust SCC mixtures.
F. Clarity
The manuscript is a paragon of clarity and organization, meticulously presenting complex analytical concepts and results in a manner that is accessible to a wide readership. Through strategic structuring and the judicious use of language, the paper demystifies the sophisticated interactions between the various components of the modified SCC. Enhanced by illustrative diagrams and tables, the narrative unfolds seamlessly, guiding the reader through the intricacies of the research with ease and fostering a deeper understanding of the dynamic properties of this novel composite material.
G. Strengths
1. The research's innovative approach to modifying SCC with a composite blend of materials stands out as a pivotal strength, offering new avenues for enhanced damping properties and sustainability.
2. The study's exhaustive experimental analysis provides a rich, detailed understanding of the modified concrete's behavior in both its fresh and hardened states, laying a solid foundation for future applications.
3. By underscoring the practical implications and environmental benefits of utilizing industrial residues in concrete, the study contributes valuable insights into sustainable construction methodologies.
4. The meticulous examination of how the chosen materials' synergistic effects influence SCC performance offers critical guidance for the advancement of concrete technology, making a compelling case for the adoption of such innovative mixtures in structural applications.
H. Potential weaknesses
1. The study's exploration, while groundbreaking, is anchored to specific combinations of additives and their concentrations, which may not encapsulate the full spectrum of possible interactions and effects on SCC properties. This focus could potentially narrow the scope of the findings and their applicability to a broader range of construction scenarios.
2. Despite the promising results, the investigation's emphasis on laboratory-based analyses may limit the direct translation of its findings to real-world applications without comprehensive field validation. This gap between theoretical potential and practical applicability necessitates further empirical research to fully harness the benefits of the modified SCC in various environmental and operational contexts.
I. Overall evaluation
Embarking on a journey to the frontiers of concrete technology, this research marks a significant stride towards understanding and optimizing the complex interactions within modified self-compacting concrete. By introducing a novel composite material that not only promises mechanical robustness and enhanced damping properties but also champions environmental sustainability, the study paves the way for future innovations in concrete design. Its contributions to the field extend beyond the mere augmentation of material properties, venturing into the realm of sustainable construction practices with far-reaching implications for structural applications. Despite of these facts, the reviewer has a lot of comments and questions to be answered.
J. Recommendations for improvement
1. Please try to answer the points of Sections H and K and incorporate all the answers in the manuscript!
2. Please broaden the spectrum of the study to encompass a wider variety of bridge types and configurations, they could significantly enhance the applicability of the findings, offering a richer understanding of the material's behavior across different contexts.
3. Please integrate additional empirical data from real-world projects and field tests; they would not only validate the theoretical models but also provide a more grounded basis for the practical application of the modified SCC, bridging the gap between laboratory research and construction practice.
4. Please do a deeper dive into the economic and logistical considerations surrounding the implementation of the recommended material combinations, they would offer a more holistic perspective on their feasibility, ensuring that the insights garnered from this study can be effectively translated into actionable strategies for sustainable infrastructure development.
5. In the Citation section on the left side of page #1, the journal name is not adequate: "Metals". Please modify it to "Materials".
6. The Reference style in the reference list is not in accordance with the requirement of the journal "Materials". Please revise all items and add DOI numbers, too!
7. The mentioned ASTM, ABNT, etc. standards are not cited accurately, they do not have items in the reference list! Please supplement them!
8. Please try to avoid using abbreviations in the Abstract!
9. The abbreviation "SCC" is not explained in Section "Introduction" in the first sentence! Please supplement!
10. Figs. 1 and 2: please use a more accurate and calibrated ruler for your photos.
11. Please list all the instruments, machines, etc. you applied in the research, give all the relevant data/details (manufacturer, type, etc.), the measurement accuracies, tolerances, calibration and validation dates!
12. If you write a reference in a figure, please use the reference numbering in rectangular bracket and the accurate reference number inside, not only the authors and the publishing year. E.g., Fig. 4. Please make a double-check in the entire manuscript!
13. Please use "–" for the negative sign (Alt+0150) instead of "-".
14. Please prepare a Nomenclature and a List of abbreviations at the end of the paper.
K. Questions
1. Given the innovative approach of incorporating rubber and marble/granite residues in self-compacting concrete (SCC), how does the long-term performance and durability of these modified SCC mixtures compare to traditional SCC formulations under various environmental conditions (e.g., freeze-thaw cycles, chemical attack, and sustained load conditions)? Please provide a comprehensive analysis, including predictive modeling and empirical data, to support the longevity and reliability of these materials in structural applications over extended periods.
2. Considering the complex interactions between the various additives (hooked-end steel fibers, rubber, marble/granite residues) in the modified SCC, can you provide a detailed microstructural characterization to elucidate how these additives influence the concrete's mechanical properties and durability? Furthermore, how might these microstructural insights inform the optimization of additive concentrations and particle size distributions to maximize performance benefits while minimizing potential drawbacks?
3. The study suggests environmental benefits from incorporating industrial residues into SCC, such as reduced cement consumption. Can you conduct a life cycle assessment (LCA) or similar quantitative analysis to evaluate the environmental impact and sustainability of using these modified SCC mixtures compared to conventional SCC and other construction materials? This analysis should consider the entire lifecycle, from raw material extraction through end-of-life recycling or disposal.
4. What are the economic implications and potential market adoption barriers for these modified SCC mixtures? Please provide an in-depth cost-benefit analysis considering the entire supply chain, from material sourcing through construction implementation. Additionally, identify and discuss any regulatory, technical, or market perception barriers that could hinder the widespread adoption of this technology in the construction industry.
5. Given the improved damping properties of the modified SCC mixtures, how do these materials perform under dynamic and cyclic loading conditions, including seismic and blast loading scenarios? Please provide a thorough evaluation using both analytical models and experimental data. The analysis should consider not only the immediate structural integrity but also the potential for energy dissipation and post-event damage mitigation, comparing the performance with that of traditional SCC and other advanced construction materials designed for high-impact resistance.
6. Considering the modified self-compacting concrete (SCC) incorporates a diverse range of additives – hooked-end steel fibers, rubber, and marble/granite residues – each with distinct physical and chemical properties, a critical question arises regarding the interfacial transition zone (ITZ) between these additives and the cement matrix. How do the varied ITZs, particularly around the steel fibers and rubber particles, affect the crack propagation resistance, fracture toughness, and overall failure mechanisms of the SCC under complex stress states (e.g., multi-axial loading and impact)? This question demands a multi-scale analysis, integrating advanced microscopy techniques (such as SEM, TEM, and Nano-CT scans) for microstructural examination with mechanical testing at the macro-scale. The analysis should not only delineate the distinct roles of each additive's ITZ in influencing the composite's performance but also explore how these micro-scale characteristics translate to macro-scale behavior, potentially informing novel mix design strategies that optimize the balance between mechanical properties and durability.
Author Response
RESPONSE TO REVIEWERS
REVISOR 1
Comments and Suggestions for Authors
- Potential weaknesses
- The study's exploration, while groundbreaking, is anchored to specific combinations of additives and their concentrations, which may not encapsulate the full spectrum of possible interactions and effects on SCC properties. This focus could potentially narrow the scope of the findings and their applicability to a broader range of construction scenarios.
AUTHORS: At first at all Authors congratulate all comments and suggestions provided. Concern to the question, the Reviewer is correct. In order to attempt this question, a new sentence was included into the revised manuscript, elucidating this limitation, as yellow highlighted (at Introduction section in the last paragraph).
Despite the promising results, the investigation's emphasis on laboratory-based analyses may limit the direct translation of its findings to real-world applications without comprehensive field validation. This gap between theoretical potential and practical applicability necessitates further empirical research to fully harness the benefits of the modified SCC in various environmental and operational contexts.
AUTHORS: The Reviewer is also absolutely correct. Considering this mentioned comment, a new sentence into Introduction section, was also included, as also yellow highlighted
- Overall evaluation
Embarking on a journey to the frontiers of concrete technology, this research marks a significant stride towards understanding and optimizing the complex interactions within modified self-compacting concrete. By introducing a novel composite material that not only promises mechanical robustness and enhanced damping properties but also champions environmental sustainability, the study paves the way for future innovations in concrete design. Its contributions to the field extend beyond the mere augmentation of material properties, venturing into the realm of sustainable construction practices with far-reaching implications for structural applications. Despite of these facts, the reviewer has a lot of comments and questions to be answered.
AUTHORS: The Reviewer is correct. Authors congratulate all comments and suggestions provided.
Recommendations for improvement- Please try to answer the points of Sections H and K and incorporate all the answers in the manuscript!
AUTHORS: Authors have majority adopted all comments and suggestions provide by this Reviewer and other two, as texts included into the revised manuscript, as yellow highlighted.
- Please broaden the spectrum of the study to encompass a wider variety of bridge types and configurations, they could significantly enhance the applicability of the findings, offering a richer understanding of the material's behavior across different contexts.
AUTHORS: This comment and suggestion is absolutely fantastic and a very good contribution is provided. However, these are modification and/or new manuscript intended to a next future paper.
- Please integrate additional empirical data from real-world projects and field tests; they would not only validate the theoretical models but also provide a more grounded basis for the practical application of the modified SCC, bridging the gap between laboratory research and construction practice.
AUTHORS: This comment and suggestion is absolutely fantastic and a very good contribution is provided. However, these are modification and/or new manuscript intended to a next future paper.
- Please do a deeper dive into the economic and logistical considerations surrounding the implementation of the recommended material combinations, they would offer a more holistic perspective on their feasibility, ensuring that the insights garnered from this study can be effectively translated into actionable strategies for sustainable infrastructure development.
AUTHORS: The Reviewer is correct, and this suggestion has a very good economical and technological contribution. Unfortunately, Authors consider that no feasible scientific novelty is provided based on these analyses. Thus, this is not incorporated. A future paper or e-book can be proposed with this Reviewer coauthoring this matter/field.
- In the Citation section on the left side of page #1, the journal name is not adequate: "Metals". Please modify it to "Materials".
AUTHORS: The modification was provided (as yellow highlighted).
- The Reference style in the reference list is not in accordance with the requirement of the journal "Materials". Please revise all items and add DOI numbers, too!
AUTHORS: This modification will be provided, if approved this manuscript, at proofing version, as suggested.
- The mentioned ASTM, ABNT, etc. standards are not cited accurately, they do not have items in the reference list! Please supplement them!
AUTHORS: The Reviewer is correct. However, it has commonly used citation of standard without requiring its complete data into list if reference. Authors will contact the Editorial office and manager/staff in order to elucidate this question and a solution will adequately be adopted.
- Please try to avoid using abbreviations in the Abstract!
AUTHORS: The Reviewer’s suggestion was adopted. All abbreviation was elucidated in its first appearance in both Abstract and Introduction.
- The abbreviation "SCC" is not explained in Section "Introduction" in the first sentence! Please supplement!
AUTHORS: The comment was adequately solved. The modifications are yellow highlighted.
- Figs. 1 and 2: please use a more accurate and calibrated ruler for your photos.
AUTHORS: The Reviewer is correct. Figs. 1 and 2 were revised and reformulated in order to clarify its scale bars. Please, see these two reworked figures.
- Please list all the instruments, machines, etc. you applied in the research, give all the relevant data/details (manufacturer, type, etc.), the measurement accuracies, tolerances, calibration and validation dates!
AUTHORS: The Reviewer is correct. These data were possibly included. Some details are not possible to be obtained. Thus, these data were not included. All modifications are also yellow highlighted.
- If you write a reference in a figure, please use the reference numbering in rectangular bracket and the accurate reference number inside, not only the authors and the publishing year. E.g., Fig. 4. Please make a double-check in the entire manuscript!
AUTHORS: The Reviewer is correct. The modifications were adopted.
- Please use "–" for the negative sign (Alt+0150) instead of "-".
AUTHORS: The Reviewer is correct. These modifications will be provided when the proof version is able.
- Please prepare a Nomenclature and a List of abbreviations at the end of the paper.
AUTHORS: The Reviewer’s suggestion is a good idea. However, Authors have decided to utilize the guideline for authors.
- Questions
AUTHORS: All questions and comments provided by tis Reviewer at section entitled “K” are excellent suggestions and ideas. Authors are planning to organize these information/data and future results to propose an e-Book or a regular book. However, these questions are not feasible to constitute a regular article. Perhaps, a review or book is a adequate local to be published.

Reviewer 2 Report
Comments and Suggestions for Authors
This study explores the effects of hooked-end steel fibers on modified self-compacting concrete (SCC), incorporating marble, granite, and rubber residues. Increased rubber content seemingly enhances energy absorption, indicating a viable structural alternative, while the combined use of materials improves the damping ratio despite slight reductions in mechanical behavior and slump flow. The study aims to offer the potential for environmentally friendly SCC designs with reduced cement consumption.
I found the topic original and relevant. Although there are many studies on the effects of additives on self-compacting concrete, this study's unique feature is the combined effects of various additives.
I believe incorporating hooked-end steel fiber into a concrete mixture is a distinctive aspect of the study.
The authors should provide the source of the microscopic images in Figures 1, 2, and 7.
Regarding Table 3, given that the superplasticizer content remains constant, my question for the authors is why the water content of the two specimens is lower since normally water and superplasticizer work together to maintain workability.
Based on Table 3, it appears that all concrete mixtures contain hooked steel fiber. If so, I wonder why the authors did not include a sample with ordinary steel fiber to justify the benefits of hooked steel fiber.
Line 208: Please relocate number (8) to the preceding line.
Line 241: RC/20SF/30MGR/5R looks like should be SSC/20SF/30MGR/5R.
Line 256: t500 looks like should be T500.
Line 256: I believe it is useful to define T500. It seemingly represents the "Flow Test."
Based on the values reported in Table 5, steel fibers apparently did not help the tensile strength specimen compared to ordinary concrete. Please clarify.
I need help determining what tests are used to estimate the compressive and tensile strength of specimens. Please clarify.
I wonder if the authors can comment on the economic aspects of replacing ordinary steel with hooked steel fibers in the concrete mixtures.
I think the references are good enough.
Author Response
REVISOR 2
This study explores the effects of hooked-end steel fibers on modified self-compacting concrete (SCC), incorporating marble, granite, and rubber residues. Increased rubber content seemingly enhances energy absorption, indicating a viable structural alternative, while the combined use of materials improves the damping ratio despite slight reductions in mechanical behavior and slump flow. The study aims to offer the potential for environmentally friendly SCC designs with reduced cement consumption. I found the topic original and relevant. Although there are many studies on the effects of additives on self-compacting concrete, this study's unique feature is the combined effects of various additives. I believe incorporating hooked-end steel fiber into a concrete mixture is a distinctive aspect of the study.
AUTHORS: Firstly, authors are immensely grateful for the suggestions and comments provided.
The authors should provide the source of the microscopic images in Figures 1, 2, and 7.
AUTHORS: The Reviewer is correct. This information is included.
Regarding Table 3, given that the superplasticizer content remains constant, my question for the authors is why the water content of the two specimens is lower since normally water and superplasticizer work together to maintain workability.
AUTHORS: The Reviewer is correct. The water-to-cement ratio is provided based on the proposed mixture of each modified SCC mixture. Thus, it can be observed that these two specimens mentioned by Reviewer have also lower cement content than other ones, i.e. 350 and 345 kg/m3, respectively. Based on this question/comment, a new sentence is included to elucidate this matter.
Based on Table 3, it appears that all concrete mixtures contain hooked steel fiber. If so, I wonder why the authors did not include a sample with ordinary steel fiber to justify the benefits of hooked steel fiber.
AUTHORS: This is an interesting question. Based on this comment, a new sentence was also included into the revised version, as yellow highlighted. Concerning to question, it is known that a concrete with ordinary (straight) steel fiber has distinctive characteristics, which would induce to equivocate comparison. For this purposed, all mixtures should also be proposed and examined. Beside, the distribution and transfer of the stresses and crack propagation propagate differently from these hooked fibers. Thus, the intercept mechanism and inhibition of crack growth reducing the likelihood of further crack propagation is different. Based on these, Authors have not included the mentioned results and its comparison. Additionally, the comparison between straight and hook end fibers not constitutes a novelty. For example, literature has recently reported the “benefits” on fresh and hardened properties of concretes containing straight and hooked fibers, as:
[a] Marcalikova Z., Cajka R., Bilek V., Bujdos D., Sucharda O. Determination of Mechanical Characteristics for Fiber-Reinforced Concrete with Straight and Hooked Fibers. Crystals. 2020;10:545. doi: 10.3390/cryst10060545.
[b] Menor A. Faris et al .Compariosn of hook and straight steel fibers additions on Malaysian fly ash-based geopolymer concrete on the slumo, density, water absorption and mechanical properties. Materials (Based). 2021; 14:1310. doi 10.3390/ma14051310.
Line 208: Please relocate number (8) to the preceding line.
AUTHORS: The suggested modification was provided. All Equation were revised and reworked when necessary.
Line 241: RC/20SF/30MGR/5R looks like should be SSC/20SF/30MGR/5R.
AUTHORS: The Reviewer is correct. The modification was made.
Line 256: t500 looks like should be T500.
AUTHORS: The suggestion was adopted.
Line 256: I believe it is useful to define T500. It seemingly represents the "Flow Test."
AUTHORS: The Reviewer is correct. The suggestion was also adopted.
Based on the values reported in Table 5, steel fibers apparently did not help the tensile strength specimen compared to ordinary concrete. Please clarify.
AUTHORS: Please, it should be observed that Table 5 depicts all samples containing steel fibers, and results corresponding with “ordinary concrete” are not provided. This, is described between lines 320 and 324 (into previous version).
I need help determining what tests are used to estimate the compressive and tensile strength of specimens. Please clarify.
AUTHORS: The Reviewer is correct. These details are included and better explained into subsection 2.3, i.e.:
“Compressive strength are carried out utilizing cylindrical specimens 100 x 200 (± 1) mm acording to ABNT NBR 5739:2018. The tensile strength is determined by using also cylindrical specimnes utilizing a dimetrical compression method (ABNT NBR 7222:2011) and modulus of elasticity by the compression (ABNT NBR 8522:2017).”
I wonder if the authors can comment on the economic aspects of replacing ordinary steel with hooked steel fibers in the concrete mixtures.
AUTHORS: The Reviewer’s point of view is very interesting. Based on this comment, new sentences were included at conclusion (item 5, yellow highlighted). However, based on our results this is impossible to be measured/calculated. This is a practice that has been commonly applied in several civil constructions. Literature has not provided an economical study (e.g. price per ton per cement ton utilized per fiber per kg) when a common straight fiber is replaced with a hooked fiber. However, it is reported that both fresh and hardened properties are considerably improved, as reported in Refs. [a] and [b]. In order to solve this question, firstly, it should be considered that there are other fibers types, e.g. flat end, undulated and hooked flat-end fiber. The differences among these are dimensions and mainly its corresponding tensile strengths, i.e. 1100, 1500 and 1200 MPa, respectively (please see https://barsandrods.arcelormittal.com/wiresolutions/steelfibre). Based on this, it is clearly perceived that flat fibers (quasi straight) have lower tensile strength than the hooked (~1350 MPa). Evidently, there exists other types reaching up to 2400 MPa (as the case of hooked glued fiber (ArcelorMittal®). The Dramix fibers utilized in this study, are also commercialized in other fiber types, i.e. 3D, 4D, etc., depending to desired construction type, e.g. flooring, underground, precast, etc. Thus, the decision to adopt a conventional fiber or a hook or other type depends strongly to desire requirements, the final application and mechanical behavior desired.
I think the references are good enough.
AUTHORS: The Authors are grateful with all comment and suggestion provided.

Reviewer 3 Report
Comments and Suggestions for Authors
The manuscript "Hooked-end steel fibers affecting damping ratio of modified self-compacting concrete with rubber and marble/granite additions" contains several inaccuracies that need to be clarified. The advantage of the manuscript is the large number of cited articles in relation to the research conducted. However, the main problem is the authors' information about the novelty of the topics discussed. The authors want to improve the quality of the environment by disposing of used tires. The manuscript shows that the addition of rubber causes a significant reduction in the basic strength parameters of SCC. Hence, the effect of using rubber in SCC does not bring benefits and actually deteriorates the properties of the basic building material, which is concrete. This needs clarification and correction in the article.
General remarks
1. Authors should write how many samples were tested in total.
2. Please write why the authors decided on such proportions of additives to SCC. Do the authors have their own thoughts, apart from literature reports?
3. Table 5 clearly shows that the addition of rubber reduces the compressive strength of concrete by 13 to 36% compared to SCC/SF/MGR concrete. So what is the point of modifying concrete with the addition of rubber?
4. It should be explained what causes such large variations in the compressive strength of SCC (336 line Fig. 4c). This needs to be clarified.
5. The conclusions contradict the results. The addition of rubber reduces the compressive strength of concrete and other mechanical parameters (conclusions 1, 2 and 3). The addition of rubber has an adverse effect on SCC (conclusion 4).
6. Applications should be supplemented with a statistical evaluation of the research.
Specific remarks
7. 125 line: Please write how many samples were actually taken (0.6?).
8. 181 line: Eq.1: write in what units ω0 and ωd are given.
9. 184, 188 line: Eq.2, 3: write in what units Z is given and what AnAn and A, r, tn mean.
10. 198 line: The n factor (from Eq.4) is missing in equation (7) This needs to be explained.
11. 208 line Eq.8: Please specify the units in which Ed, Es are expressed.
12. 338 line: It is natural that when the presence of a material with a much higher stiffness increases, the stiffness of the composite also increases.
13. 366 line, Fig 4c: Why is the SD for SCC tensile strength not shown?
14. Fig.6: The damping symbol δ or ζ should be unified?
I recommend an in-depth review of the manuscript, including comments, to make it an article suitable for publication in the Materials.
In its current state, the article should not be published.
Comments on the Quality of English LanguageMinor editing of English language required.
Author Response
REVISOR 3
The manuscript "Hooked-end steel fibers affecting damping ratio of modified self-compacting concrete with rubber and marble/granite additions" contains several inaccuracies that need to be clarified. The advantage of the manuscript is the large number of cited articles in relation to the research conducted. However, the main problem is the authors' information about the novelty of the topics discussed. The authors want to improve the quality of the environment by disposing of used tires. The manuscript shows that the addition of rubber causes a significant reduction in the basic strength parameters of SCC. Hence, the effect of using rubber in SCC does not bring benefits and actually deteriorates the properties of the basic building material, which is concrete. This needs clarification and correction in the article.
General remarks
- Authors should write how many samples were tested in total.
AUTHORS: The Reviewer is correct. This information was included. The modifications are yellow highlighted.
“In order to guarantee reproducibility and to determine the compressive strength, for each one of the proposed mixtures, at least 6 specimens are molded and tested at 7 and 28 days of age. Thus, the number of the used specimens to conduct the compression, tensile and modulus of elastic measurements are 10, 4 and 4, respectively. This totalizes 60 + 24 + 24 = 108 specimens considering all proposed mixtures.”
- Please write why the authors decided on such proportions of additives to SCC. Do the authors have their own thoughts, apart from literature reports?
AUTHORS: The Reviewer is correct. There exists a sentence into the manuscript describing “The selection of mixes is based on dosage studies for SCC developed by [11,12,49-50].”. However, these mixtures can also be compared with other previous articles, e.g. [33-40] and [61-65]. This information was also included into the revised version of the manuscript.
- Table 5 clearly shows that the addition of rubber reduces the compressive strength of concrete by 13 to 36% compared to SCC/SF/MGR concrete. So what is the point of modifying concrete with the addition of rubber?
AUTHORS: The Reviewer is correct. This is also a good question. Although it is recognized that rubber also eventually decreases compressive behavior, there is a remained speculation to certain improvement in tensile aspect be positively modified. It was based on this perspective (previously reported) that rubber content was included and considered to prepare other mixtures containing rubber contents. This information was also included into the revised version of the manuscript, as yellow highlighted.
- It should be explained what causes such large variations in the compressive strength of SCC (336 line Fig. 4c). This needs to be clarified.
AUTHORS: Considering this question, the Reviewer’s comment is probably confused or equivocate. This, due to Fig.4c depicts tensile-to-compressive ratio. At line 336, Table 6 is demonstrated, and no compressive strengths are reported.
- The conclusions contradict the results. The addition of rubber reduces the compressive strength of concrete and other mechanical parameters (conclusions 1, 2 and 3). The addition of rubber has an adverse effect on SCC (conclusion 4).
AUTHORS: The Reviewer is correct when rubber content decreases the resulting compressive strength. However, Authors have not consider that contradict is present. Based on the comment, conclusion was revised.
- Applications should be supplemented with a statistical evaluation of the research.
AUTHORS: The Reviewer is correct. It is included that error ranges are occurring in 10% of absolute values (texts in yellow highlighted). No chi-squared or coefficient of variation were determined. This is based on other previous published articles.
Specific remarks
- 125 line: Please write how many samples were actually taken (0.6?).
AUTHORS: The Reviewer is correct. The text was revised and reworked. The modifications are yellow highlighted.
- 181 line: Eq.1: write in what units ω0and ωdare given.
AUTHORS: The Reviewer is correct. Modifications are yellow highlighted.
- 184, 188 line: Eq.2, 3: write in what units Z is given and what AnAnand A, r, tnmean.
AUTHORS: The Reviewer is correct. The text was modified.
- 198 line: The n factor (from Eq.4) is missing in equation (7) This needs to be explained.
AUTHORS: The Reviewer is correct. Corrections were provided.
- 208 line Eq.8: Please specify the units in which Ed, Esare expressed.
AUTHORS: The Reviewer is correct. Corrections were provided
- 338 line: It is natural that when the presence of a material with a much higher stiffness increases, the stiffness of the composite also increases.
AUTHORS: The Reviewer is correct. However, no stiffness results are discussed.
- 366 line, Fig 4c: Why is the SD for SCC tensile strength not shown?
AUTHORS: As previously commented and rebutted, Fig. 4c depicts a tensile-to-compressive ratio. The SD for both compressive and tensile strengths are demonstrated into Table 5 (at line 271).
- Fig.6: The damping symbol δ or ζ should be unified?
AUTHORS: The Reviewer is correct. The modifications were provided and are yellow highlighted.
I recommend an in-depth review of the manuscript, including comments, to make it an article suitable for publication in the Materials.
AUTHORS: Authors are very grateful with all comment and suggestions provided. We have intensively worked to revise and improve the manuscript. Based on these improvements, it is hoped that the manuscript deserves its final publication.
_ _ __

Round 2
Reviewer 1 Report
Comments and Suggestions for Authors
Dear Authors.
many thanks for your valuable revision.
There are two missing things:
1. You have not answered any of my questions (see the previous review, Section K).
2. Please prepare a Nomenclature and a List of abbreviations at the end of the paper. – I cannot accept your given answer. Without these sections, your manuscript would have a significant deficit. This kind of valuable study with many abbreviations and symbols must have a Nomenclature and a List of abbreviations.
Yours sincerely,
The Reviewer
Author Response
RESPONSE TO REVIEWERS
REVISOR 1 (Round 2)
Dear Authors.
many thanks for your valuable revision.
There are two missing things:
- You have not answered any of my questions (see the previous review, Section K).
AUTHORS: Dear Reviewer, firstly we again congratulate your question, comments and suggestions provided. Secondly, we also apologize to not elucidate the reason for no responses considering those questions at section K. At 1st round, we have considered those are questions potentially to be solved in future works, which not decrease the scientific merit of the present manuscript (QUESTIONS #1, 2 and 3). For instance, it is suggested at item #1 that predictive modeling and empirical data be included. It is induced that this no substantially depreciates the scientifically merit and will taken more than 24 months (e.g. evaluate the entire lifecycle) to be developed into a Master or Doctoral studies. On the other hand, there are other items, which will taken a long time period to be attained, e.g. to provide an in-depth cost-benefit analysis considering the entire supply chain, from material sourcing through construction implementation (QUESTION #4). Besides, this is strongly dependent of the desired parameters in a specific civil construction and the economy of the country in question. Another point of view is the damping properties including seismic and blast loading scenarios. This is a great and wonderful suggestion. However, in a regular paper, it is considered that it is “almost” impossible to determine in short-time period (QUESTION #5). Finally, the QUESTION #6 is also at same line of that Question #1, which demands a long-time to be solved and this can potentially be considered in future papers and e-books optionally.
- Please prepare a Nomenclature and a List of abbreviations at the end of the paper. – I cannot accept your given answer. Without these sections, your manuscript would have a significant deficit. This kind of valuable study with many abbreviations and symbols must have a Nomenclature and a List of abbreviations.
AUTHORS: Authors again apologize for this question. Based on this, without knowledge into guidelines to authors, two sections with list of both abbreviation and symbols are proposed, as yellow highlighted.
Yours sincerely,
The Reviewer
REVISOR 3 (Round 2)
Compared to the previous version of the article, they have introduced corrections that partially take into account the reviewer's suggestions. However, there is one more inconsistency.
Re 3: Which does not change the fact that the addition of rubber significantly worsens the compressive strength of concrete. After taking into account the above, the article may be published.
AUTHORS: Firstly, Authors again congratulate the comments provided. Considering this new suggestion, it is stated the Reviewer is completely correct. Based on this, a new elucidating this fact is included, as yellow highlighted. This is included at Conclusion #3 and at Discussion section at page 18/25.

Reviewer 3 Report
Comments and Suggestions for Authors
Compared to the previous version of the article, they have introduced corrections that partially take into account the reviewer's suggestions. However, there is one more inconsistency.
Re 3: Which does not change the fact that the addition of rubber significantly worsens the compressive strength of concrete.
After taking into account the above, the article may be published.
Comments on the Quality of English LanguageMinor editing of English language required.
Author Response

(The authors gave the same response as above.)
